# Immune Checkpoint Inhibitor-Induced Pancreatic Injury (ICI-PI) in Adult Cancer Patients: A Systematic Review and Meta-Analysis

**DOI:** 10.3390/cancers17071080

**Published:** 2025-03-24

**Authors:** Cha Len Lee, Israt Jahan Riya, Ifrat Jahan Piya, Thiago Pimentel Muniz, Marcus Otho Butler, Samuel David Saibil

**Affiliations:** 1Department of Medical Oncology and Hematology, CancerCare Manitoba, University of Manitoba, Winnipeg, MB R2H 2A6, Canada; 2Division of Medical Oncology and Hematology, Princess Margaret Cancer Center, University of Toronto, University Health Network, Toronto, ON M5S 1Z5, Canadasam.saibil@uhn.ca (S.D.S.); 3Dhaka Medical College Hospital, Dhaka 1000, Bangladesh

**Keywords:** immune checkpoint inhibitors, ICI pancreatitis, immune-related pancreatitis, diabetes mellitus, exocrine pancreatic insufficiency

## Abstract

This study looked at the frequency of pancreatitis, a type of inflammation in the pancreas, in patients who receive immune checkpoint inhibitors (ICIs) for cancer treatment, along with how it is managed, and the outcomes. We found that while this condition is rare, it can lead to long-term health issues like diabetes mellitus and exocrine pancreatic insufficiency, even after cancer treatment ends. Our research also points out several missing pieces of evidence regarding clear diagnostic guidelines, better treatment plans, and new ways to predict who might be at risk. This can help to make immunotherapy safer and more effective for patients.

## 1. Introduction

Immune checkpoint inhibitors (ICIs) have revolutionized the field of cancer therapeutics by improving survival outcomes across various malignancies [1]. These agents target immune checkpoints, including programmed cell death protein 1 (PD1), ligand PDL1, and cytotoxic T-lymphocyte antigen 4 (CTLA4), thereby enhancing T-cell-mediated antitumor responses [2]. Despite their success, ICIs are associated with a distinctive spectrum of immune-related adverse events (irAEs) due to immune system overactivation [3]. Unlike conventional chemotherapy toxicities, irAEs can affect any organ system, have a delayed onset, and may result in long-term morbidities [4].

ICI-induced pancreatic injury (ICI-PI), also known as type 3 autoimmune pancreatitis, is characterized by the T-cell-mediated destruction of pancreatic β-cell, reduced pancreatic function, and pancreatic atrophy [5,6,7]. The pathophysiology of ICI-PI involves T-cell activation triggered by ICIs, leading to pancreatic injury. Clinical studies have documented the increased infiltration and activation of cytotoxic T-cells within the pancreas in patients with ICI-PI [8]. The incidence of ICI-PI is estimated at 1–2%, with higher rates observed in patients receiving dual ICI regimens or concurrent systemic therapies [9]. The onset varies widely, ranging from the initial immunotherapy cycles to over a year later, with a median onset of nine months [10]. Clinical presentations can span from asymptomatic hyperlipasemia to more typical clinical manifestations, such as epigastric pain, fever, nausea, vomiting, and diarrhea [11,12,13]. The diagnostic criteria for ICI-PI are also broad, ranging from asymptomatic biochemical abnormalities to visible pancreatic inflammation detectable via radiological imaging. The diagnosis requires the exclusion of other common causes such as alcohol use, hypertriglyceridemia, bile stones, autoimmune pancreatitis, pancreatic metastases, or non-ICI medications. The severity of ICI-PI is not determined by the numerical elevation of lipase or amylase levels in laboratory tests; instead, it is graded according to the Common Terminology Criteria for Adverse Events (CTCAE) versions 4.0 or 5.0. Grade 2 ICI-PI is characterized by enzyme elevation or radiological findings, and grade 3 is defined by the presence of clinical symptoms that necessitate medical intervention [14,15].

A patient may present with a grade 3 hyperlipasemia yet still be classified as having grade 2 ICI-PI if they remain asymptomatic [16]. Physicians frequently encounter asymptomatic hyperlipasemia. This poses a diagnostic challenge that complicates treatment decisions due to the unclear clinical significance of pancreatic enzyme elevations without accompanying symptoms. While a watch-and-wait approach is often considered safe, the long-term implications of asymptomatic hyperlipasemia on pancreatic function remain uncertain. This challenge is compounded by the absence of standardized guidelines for managing ICI-PI [17,18]. Furthermore, the association between irAEs and survival benefits complicates decision-making regarding the early use of steroids, alternative immunosuppressants, or the permanent cessation of ICI therapy, as these interventions are controversial due to their potential impact on survival outcomes [19,20,21,22,23,24].

From our observation, studies on ICI-related pancreatic toxicities have not thoroughly investigated the aspect of the long-term consequences in patients with asymptomatic hyperlipasemia [25,26,27,28]. This represents a significant gap in understanding of the progression and chronic adverse complications, such as chronic pancreatitis, pancreatic atrophy, diabetes mellitus, and exocrine pancreatic insufficiency. As these problems could develop in the absence of symptoms, long-term monitoring is warranted even after the discontinuation of ICI therapy [29]. This systematic review and meta-analysis aims to address these gaps in ICI-PI by synthesizing data on its frequency, severity, serum abnormalities, treatment approaches, long-term outcomes, and associated adverse complications. Additionally, we explore the occurrence of concurrent non-pancreatic irAEs to provide a more comprehensive understanding of this immunotoxicity.

## 2. Materials and Methods

### 2.1. Search Strategy and Protocol

This systematic review and meta-analysis was conducted according to the Preferred Reporting Items for Systematic Reviews and Meta-analyses (PRISMA) and the Cochrane Handbook for Systematic Reviews of Interventions, as shown in Appendix A [30,31]. We registered the protocol in PROSPERO under registration number CRD42024583775.

A systematic search of the PubMed, EMBASE, and Cochrane Library databases identified relevant studies from 2010 to 8 August 2024. The search strategy is detailed in Appendix A. Three authors (CLL, IJR, and IJP) independently screened the search results on Rayyan and included studies based on pre-determined inclusion and exclusion criteria [32].

### 2.2. Study Selection

We included prospective and retrospective studies that met the criteria based on the population, intervention, control, and outcomes (PICO) framework. Eligible studies reported ICI-PI, defined as a spectrum ranging from asymptomatic hyperlipasemia to symptomatic pancreatic inflammation following ICI treatment, characterized by laboratory and radiological abnormalities in the absence of alternative etiologies. ICI-PI severity was graded using the Common Terminology Criteria for Adverse Events (CTCAE) versions 4.0 or 5.0, as specified in each study [14,15]. Mild to moderate ICI-PI (grade ≤ 2) was defined by the presence of pancreatic enzyme elevation or radiological findings, while severe cases (grade ≥ 3) were characterized by clinical symptoms requiring medical intervention [14,15]. ICIs included PD1 inhibitors (pembrolizumab, nivolumab), PDL1 inhibitors (atezolizumab, avelumab, durvalumab), and CTLA4 inhibitors (ipilimumab). The combination ICI therapy was a CTLA4 inhibitor plus a PD1 or PDL1 inhibitor.

The outcomes of interest included the frequency and grading of ICI-PI, tumor category, ICI modality, concurrent irAEs, serum and radiologic abnormalities, clinical management strategies, and chronic complications such as diabetes mellitus (DM) and exocrine pancreatic insufficiency (EPI). Case reports, case series with ≤5 cases, registry studies, preclinical studies, and investigations involving non-ICI immunotherapies were excluded.

### 2.3. Data Extraction

Data extraction was independently performed by three authors (CLL, IJR, and IJP), with any disagreements resolved through consensus. Extracted data included patient demographics, tumor characteristics, ICI regimens, frequency and severity of ICI-PI, lipase elevations, management strategies, chronic complications, and outcomes, including objective response rate (ORR), progression-free survival (PFS) and overall survival (OS).

### 2.4. Methodological Quality

The quality of the included studies was evaluated using the JBI critical appraisal tool, which assesses methodological rigor through 11 criteria. Detailed results are presented in Appendix A [33].

### 2.5. Data Analyses

All statistical analyses were conducted using R software (version 4.0.0; R Core Team 2024, Vienna, Austria). A random effects model was employed for meta-analysis to pool the proportions with 95% confidence intervals (CI). For outcomes where individual study proportions were <0.2 or >0.8, the logit transformation was applied, while the double arcsine transformation was used for proportions of 0 or 1 [34,35]. Heterogeneity across studies was assessed using the Cochrane Q test and the I^2^ statistics. Heterogeneity was considered significant if the *p*-value was <0.1 and the I^2^ exceeded 30%.

Studies with full recruitment were excluded in the subgroup pooled analyses to avoid overestimations. Subgroup analyses were conducted to evaluate the frequency of ICI-PI based on monotherapy versus combination ICI regimens. Leave-one-out sensitivity analyses were conducted for outcomes demonstrating significant heterogeneity, where one study was omitted at a time to reassess the results. Publication bias was assessed both visually using funnel plots and quantitatively using Egger’s test, the latter being applied when more than ten studies were included in the outcome of interest.

## 3. Results

### 3.1. Patient Selection

In this meta-analysis, 77 eligible articles were identified. After excluding duplicates and eligibility assessment, 25 retrospective studies involving a total of 48,704 patients were included in this meta-analysis (Figure 1) [10,11,36,37,38,39,40,41,42,43,44,45,46,47,48,49,50,51,52,53,54,55,56,57,58]. The reasons for study exclusion are described in Appendix A.

### 3.2. Study Characteristics

The study population comprised patients with thoracic/head and neck malignancies (38%), melanoma and skin cancers (26%), genitourinary or gynecological (18%), gastrointestinal (12%), and other malignancies (6%). The median age of participants ranged from 56 to 73 years, and 14.8% (48/324) had a history of pre-existing diabetes. The baseline patient characteristics are summarized in Table 1. Details regarding tumor site, stage, and median time to resolution are provided in Table 2. The median follow-up period varied from 2.5 to 45.9 months.

The pooled frequency of ICI-PI was 12.18% (997/48,704; 95% CI: 2.93–26.50%), with grade 1–2 toxicities accounting for 40.55% (206/461; 95% CI: 18.63–64.68%) and grade ≥3 toxicities comprising 59.45% (255/461; 95% CI: 35.32–81.37%) of cases, as illustrated in Figure 2. When the three studies with 100% ICI-PI patient recruitment were excluded, the pooled frequency decreased to 3.60% (835/48,542; 95% CI: 1.64–6.28%) [38,42,44]. Subgroup analysis demonstrated that combination ICI therapy exhibited the highest rate of ICI-PI at 7.44%, compared to anti-PD1/PDL1 monotherapy with 5.01% and CTLA4 monotherapy with 1.99% (*p* for subgroup differences <0.01) (Figure 3). The overall frequency of any-grade hyperlipasemia was reported in 84.51% (598/730; 95% CI: 61.35–98.10%) of patients (Figure 2B). The median time to onset of ICI-PI ranged from 30 to 390 days following treatment initiation.

Concurrent irAEs affecting other organs were reported in 78.85% of patients with ICI-PI (197/285; 95% CI: 59.32–93.18%), as shown in Appendix A. These included endocrine irAEs in 17.09% (39/285; 95% CI: 8.20–28.40%) of patients, gastrointestinal in 16.30% (53/285; 95% CI: 9.69–24.23%), hepatic in 11.76% (37/285; 95% CI: 4.13–22.60%), skin in 8.53% (28/285; 95% CI: 4.75–13.29%), pulmonary in 6.05% (19/285; 95% CI: 3.58–9.11%), and other organ involvement in 6.30% (21/285; 95% CI: 2.92–10.85%).

### 3.3. Chronic Adverse Outcomes of ICI-PI

Chronic adverse outcomes were reported in 63.54% of ICI-PI patients (215/561; 95% CI: 29.03–91.56%), with DM occurring in 89.45% (183/215; 95%: 61.88–100.0%) and exocrine pancreatic insufficiency (EPI) in 10.55% (32/215; 95%: 0.0–38.12%) (Table 3 and Figure 4). Among patients with DM, 80.07% (18/22; 95% CI: 24.49–100.0%) required insulin therapy, while 19.93% (4/22; 95% CI: 0.0–75.51%) were managed with oral hypoglycemic agents (Appendix A).

Among the thirty-two patients diagnosed with EPI, five were diagnosed through fecal elastase-1 levels, one through fecal fat analysis, and the remaining cases were diagnosed based on clinical symptoms [36,39,41,53].

### 3.4. Clinical Management of ICI-PI

We analyzed the pooled proportions for the management of ICI-PI based on available data. Oral and/or intravenous corticosteroids were administered to 30.20% of patients (129/547; 95% CI: 15.84–46.89%), intravenous fluids were given to 22.82% (72/293; 95% CI: 7.71–42.96%), and acute hospitalization was required in 30.46% (58/211; 95% CI: 15.43–48.03%) of cases (Figure 5). One study reported the utilization of immunosuppressants in six patients: mycophenolate mofetil (n = 3), extracorporeal photopheresis (n = 2), and intravenous immunoglobulin (n = 1) [52]. Two studies reported a median time to resolution of ICI-PI ranging from 55 to 84 days, either spontaneously or with the intervention [11,48].

Among the patients diagnosed with ICI-PI, 30.51% (163/507; 95% CI: 18.94–43.49%) required the permanent discontinuation of ICIs, while 44.17% (109/350; 95% CI: 21.96–67.68%) were able to resume therapy, as shown in Appendix A. Notably, 27.16% (25/90; 95% CI: 8.95–50.72%) of those who resumed ICI therapy experienced ICI-PI recurrence. In terms of radiological evaluations, imaging was completed in 74.71% of patients with ICI-PI (295/624; 95% CI: 40.62–97.0%), with computed tomography (CT) scans utilized in 57.87% (118/416; 95% CI: 9.99–97.35%) of cases (Figure 5D).

### 3.5. Survival Endpoints

There were insufficient data for a pooled analysis of OS and PFS. The available reported median OS ranged from 20.1 to 21.8 months [39,46,48]. Most studies did not provide data specific to patients with ICI-PI. Only one study evaluated OS with corticosteroid use among ICI-PI patients, reporting a hazard ratio (HR) of 0.73 (95% CI: 0.34–1.55) [11]. No efficacy data on OS with immunosuppressant use were available.

The ORR specific to ICI-PI patients was 61.73% (130/211; 95% CI: 55.08–68.17%), as shown in Appendix A.

### 3.6. Sensitivity Analysis and Bias Assessment

Sensitivity analysis using a leave-one-out test was conducted to identify the influential studies (Appendix A). Publication bias was evaluated using funnel plots and Egger’s test.

In assessing the frequency of ICI-PI using a leave-one-out sensitivity analysis, no single study was identified as contributing to the pooled overall heterogeneity estimate [40]. Funnel plot asymmetry and Egger’s test indicated a potential publication bias (9.27; 95% CI: 4.69–13.85; t = 3.969; *p* = 0.0006).

All the studies included were retrospective. Using the JBI tool for cohort studies to assess the risk of bias, all studies were classified as having a moderate to low risk of bias, likely due to their retrospective nature. No study fulfilled all quality criteria; however, all scored above 5 points, with none scoring below 4 (Appendix A).

## 4. Discussion

This single-arm meta-analysis, comprising 48,704 patients from 25 retrospective studies, evaluates the characteristics, frequency, and chronic complications of patients diagnosed with ICI-PI. Our main findings are as follows: First, ICI-PI is a rare immunotoxicity with a pooled incidence of 3.60%, with severe cases (grade ≥ 3) accounting for 59.45%. Second, the frequency of ICI-PI appears to be higher with combination ICI therapies. Third, hyperlipasemia was observed in 84.51% of cases, often without overt clinical symptoms. Fourth, ICI-PI was associated with chronic complications in 63.45% of cases, predominantly ICI-induced DM (89.45%) and EPI (10.55%). Lastly, 30.20% and 30.46% of ICI-PI patients received corticosteroids and acute hospitalization, respectively.

Our analysis demonstrated a higher frequency rate and grade of ICI-PI compared to a prior meta-analysis [25]. This variability may be attributed to population selection and the diagnostic criteria used for pancreatitis. We noted a high rate of hyperlipasemia due to the diverse definitions of ICI-PI across the included studies [25]. Since ICI-PI accounted for 71% of hyperlipasemia cases, while the remainder were attributable to other causes unrelated to ICI drugs, it was reasonable to diagnose ICI-PI based on lipase level abnormalities, despite the absence of other clinical signs or symptoms [36]. We included patients with diverse solid malignancies who experienced ICI-PI, leading to a heterogeneous study population in terms of cancer types, stages, treatment regimes, and comorbidities. Although we intended to isolate ICI-PI as being solely due to ICIs, the influence of prior or subsequent therapies such as chemotherapy or radiotherapy could not be fully excluded and may have contributed to the overestimated values. Ngamphaiboon et al. reported four cases of pancreatic cancer in their study, while no cases were observed in other studies [48]. Due to the small number of pancreatic cancer cases, it was not possible to establish a clear direct relationship between ICI-PI complications and pancreatic cancer in our analysis. Our study predominantly consisted of lung, melanoma, and genitourinary cancers, where the standard of care consists of dual ICI therapies which are associated with a higher risk of ICI-PI [27,28,40,44]. As shown in our subgroup analysis, the dual CTLA4 with PD1 inhibitors had the highest frequency rate of ICI-PI (7.44%; 95% CI: 4.87–10.52%) compared to monotherapy PD(L)1 inhibitors (5.01%; 95% CI: 0.80–12.52%) and CTLA4 inhibitors (1.99%; 95% CI: 1.06–3.21%). Longer treatment durations were also associated with a higher prevalence of ICI-PI [59,60]. The inclusion of late-onset cases in our study, based on the median time to onset, may have further inflated the frequency rates of ICI-PI and concurrent irAEs, which were reported to be 78.85%.

Pancreatic immunotoxicity does not appear to impact patient survival [61,62]. In our analysis, the ORR in ICI-PI patients was 61.73%. Yet, it can result in life-long pancreatic endocrine and exocrine dysfunctions such as DM and EPI [18]. ICI-induced DM is the most common long-term complication, with reported incidence rates from 0.86% to 1.27% among patients exposed to ICI agents [12,61,63]. Again, these figures varied across studies due to differences in the inclusion criteria. Our analysis demonstrated a comparatively high prevalence of both DM and EPI, with no evidence of significant publication bias. Over half of ICI-induced DM cases can present with abrupt-onset diabetic ketoacidosis, a medical emergency necessitating acute hospitalization for insulin therapy, electrolyte correction, and fluid replacement [5], whereas 4.3% of ICI-PI cases progress to EPI, which requires life-long pancreatic enzyme replacement. Common heralding presentations include hyperglycemia, fatigue, headache, nausea, abdominal discomfort, weight loss ( > 5%), and steatorrhea [39]. The key takeaway is that ICI-DM and EPI are irreversible, life-altering conditions requiring early recognition, timely intervention, and vigilant monitoring of pancreatic function, even after discontinuing ICI therapy. Given the potential complications associated with ICI-PI, it may be reasonable to consider a short course of corticosteroids in asymptomatic cases, particularly in those with persistent hyperlipasemia, despite ICI therapy being on hold. As other-organ irAEs frequently co-occur in patients with ICI-PI, with an incidence of 36%, healthcare providers need to be proactive in implementing monitoring and aggressive management [40].

Corticosteroids were administered to over a third of our study population for ICI-PI management, which was lower than previously reported [25]. Most studies have evaluated steroid use across a broad range of irAEs rather than being specific to ICI-PI. Current NCCN guidelines recommend prednisone or intravenous methylprednisolone at doses of 1–2 mg/kg/day for severe or life-threatening ICI-PI, with a tapering period of 4–6 weeks [64]. For CTCAE grade 2 pancreatitis, lower corticosteroid doses (0.5–1.0 mg/kg/day) may be considered in certain cases, although this is not explicitly defined and is entirely dependent on the physician’s judgment. These recommendations align with the American Society of Clinical Oncology (ASCO) guidelines, while the European Society for Medical Oncology (ESMO) does not provide specific guidance on corticosteroid use for ICI-PI [17,18]. In steroid-refractory cases, where a patient’s condition or imaging fails to improve within 48 to 72 h of initiating steroid therapy, second-line non-steroidal immunosuppressants have been employed. Varnier et al. highlighted an increasing trend since 2018 in the use of organ-specific immunosuppressants to minimize steroid use in managing irAEs [65]. This shift has been driven by reports indicating a negative association between corticosteroid use and antitumor outcomes, potentially explaining the lower corticosteroid usage rate in our analysis [19,21,22]. Like other studies, our outcome analysis regarding non-steroidal immunosuppressants was limited, with only six cases identified. Therefore, we were unable to draw firm conclusions about their benefits compared to corticosteroids in treating ICI-PI.

### Strengths and Limitations

The primary strength of our study is the inclusion of real-world data, which provides valuable insights into the long-term implications of this rare immunotoxicity. The reliance on retrospective studies introduces potential biases due to variability in treatment history reporting, inconsistencies in corticosteroid usage documentation, and differences in the diagnostic methods for ICI-DM and EPI. These inevitably limit our ability to fully evaluate corticosteroids in terms of their dosing, timing, OS, and long-term benefits. These limitations highlight avenues for future investigations in ICI-PI patients, particularly concerning predisposing risk factors. For example, pre-existing diabetes is shown to be a significant risk factor for ICI-PI complications, with an odds ratio of 5.91 (95% CI, 3.34–10.45) [59,61]. In our study, 14.8% of the population had pre-existing diabetes before ICI therapy. However, disentangling the diabetogenic effects of steroid treatment from the underlying diabetes remains challenging and is, frankly, inadequate based on an analysis of retrospective multicenter data. Additionally, genetic predispositions, including the presence of human leukocyte antigen (HLA-DR4) and glutamic acid decarboxylase antibodies (GADA), may increase susceptibility to ICI-PI but are not well-studied yet [10,57,58]. This suggests a need for further research on risk biomarkers to elucidate the underlying mechanisms to allow the early detection and development of targeted interventions aimed at mitigating long-term endocrine sequelae. Future research should also explore the pancreatic safety profiles of newer immunotherapeutics, such as anti-LAG3 and TIGIT inhibitors, given their emerging roles in the treatment of various malignancies.

## 5. Conclusions

Our study confirmed the prevalence of long-term complications associated with ICI-PI, such as ICI-induced DM and EPI, emphasizing the importance of early recognition, timely intervention, and vigilant monitoring of pancreatic functions, even after discontinuing ICI treatment. Despite these findings, the heterogeneity of our results limits definitive treatment guidance for patients with isolated, asymptomatic pancreatic enzyme abnormalities. This highlights the need for an adaptive research approach, utilizing registry-based studies with patient-centric endpoints that allow for data reproducibility, and a more nuanced understanding of treatment effects in such under-researched conditions, which are not easily amenable to traditional clinical trial models. Other key research gaps persist in establishing standardized diagnostic criteria, preventive strategies, and predictive markers, all of which are critical for improving patient care and outcomes in the evolving field of immuno-oncology.

## Figures and Tables

**Figure 1 cancers-17-01080-f001:**
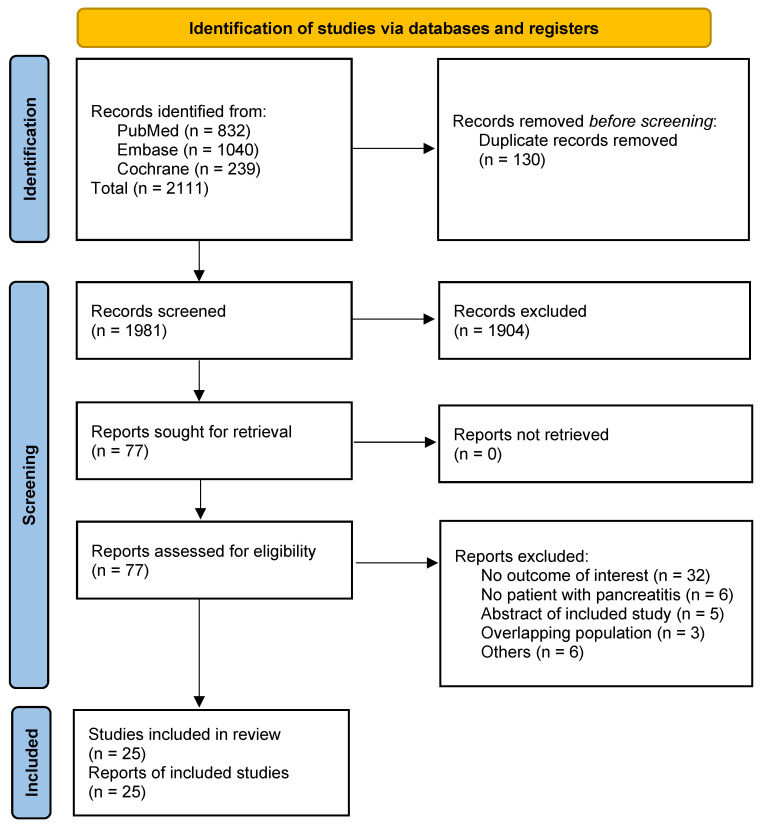
The PRISMA flow diagram for study screening and selection.

**Figure 2 cancers-17-01080-f002:**
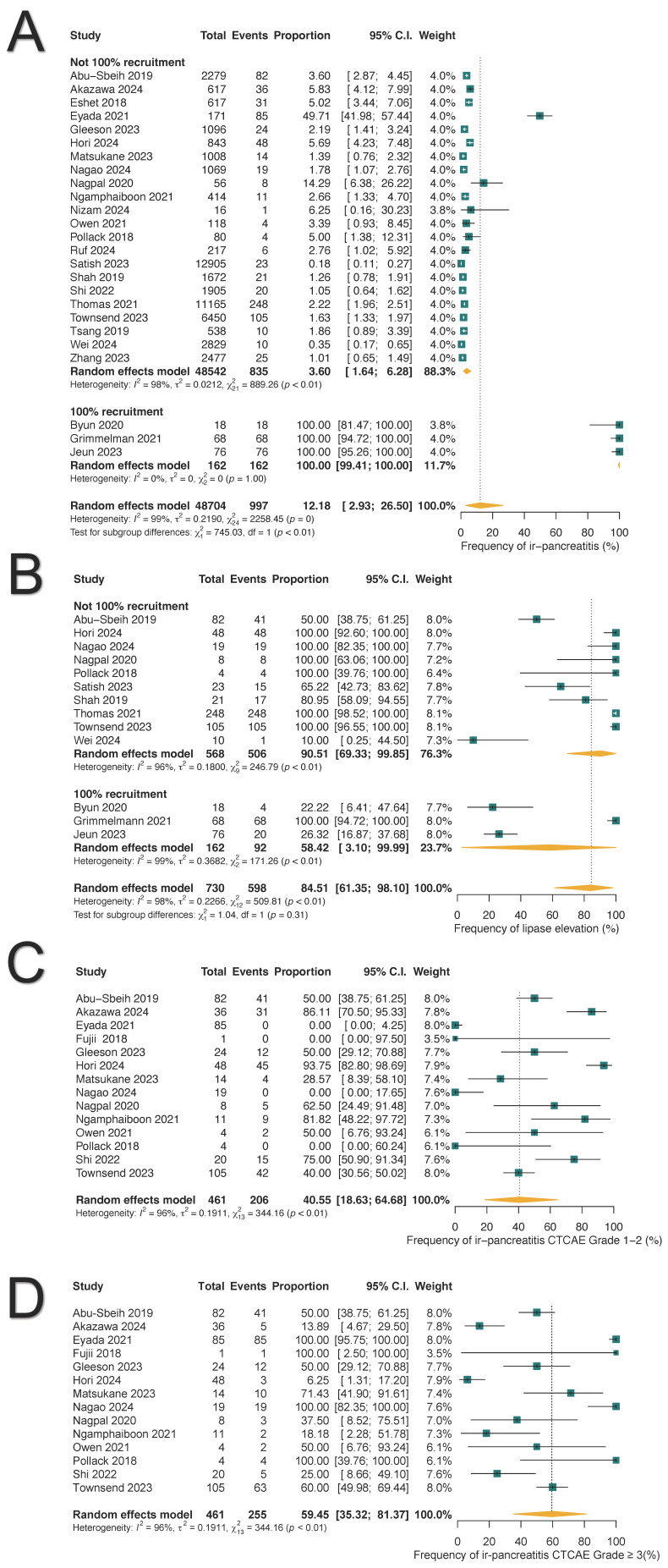
Forest plots illustrate the frequency, toxicity grades, and hyperlipasemia associated with immune-related pancreatitis (ir-pancreatitis): (**A**) Pooled frequency of all-grade ir-pancreatitis. (**B**) Pooled frequency of all-grade hyperlipasemia. (**C**) Frequency of grade 1–2 ir-pancreatitis. (**D**) Frequency of grade ≥ 3 ir-pancreatitis.

**Figure 3 cancers-17-01080-f003:**
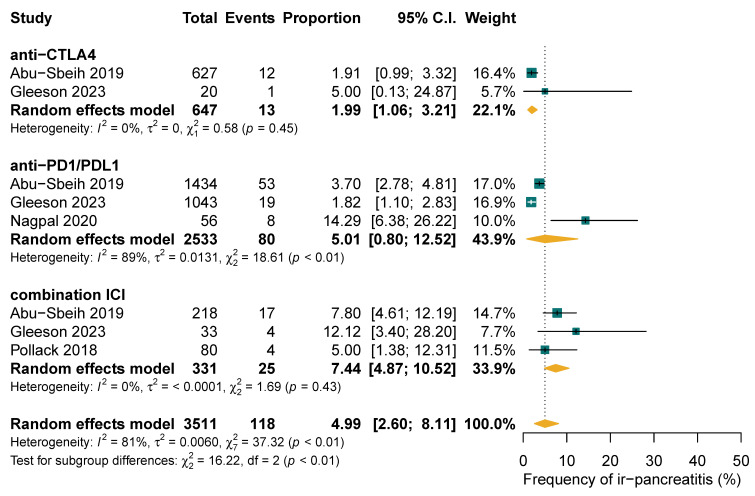
Subgroup analysis of frequency of immune-related pancreatitis (ir-pancreatitis) comparing single-agent versus combination treatment regimens.

**Figure 4 cancers-17-01080-f004:**
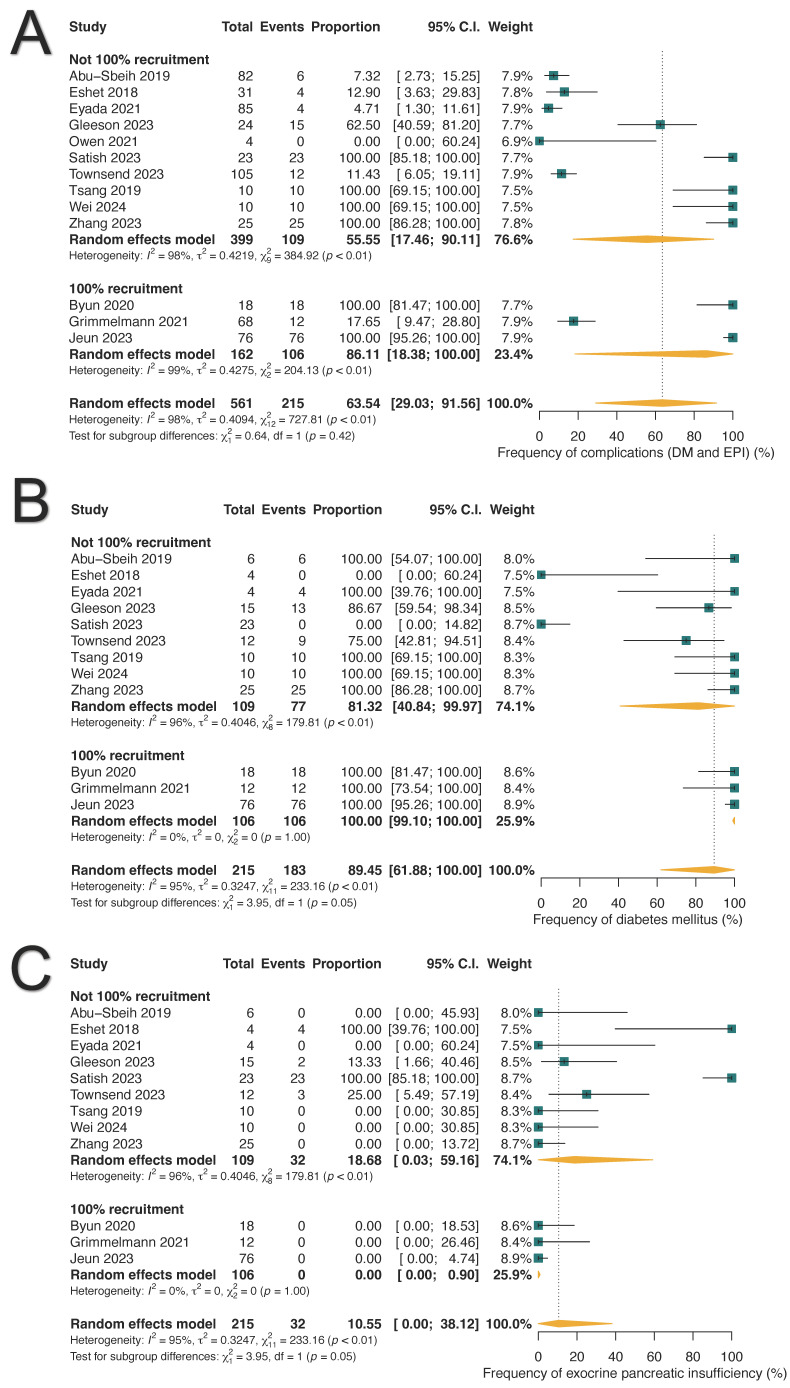
Long-term pancreatic complications associated with immune-related pancreatitis. (**A**) Pooled frequency of long-term complications. (**B**) Prevalence of diabetes mellitus (DM). (**C**) Prevalence of exocrine pancreatic insufficiency (EPI).

**Figure 5 cancers-17-01080-f005:**
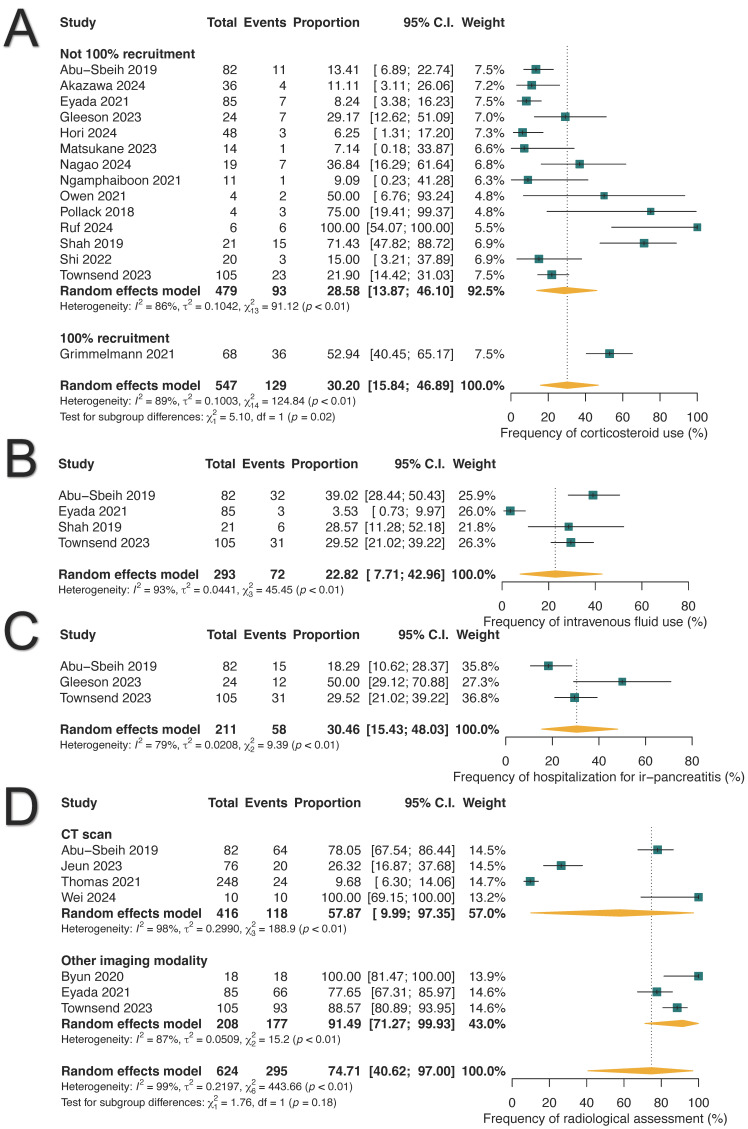
Management strategies for immune-related pancreatitis (ir-pancreatitis). (**A**) Frequency of corticosteroid use. (**B**) Frequency of intravenous fluid administration. (**C**) Frequency of acute hospitalization. (**D**) Frequency of radiological assessment.

**Table 1 cancers-17-01080-t001:** Baseline characteristics of the studies included in the analysis.

Study ^$^	Total Patients	Patients with ICI-PI (n/%)	Male (n/%)	Median Age; Years (IQR or Range)	Median Follow-Up; Months (Range)
Abu-Sbeih, 2019 [11]	2279	82/3.6	54/65.9	57 (14) ^†^	NA
Akazawa, 2014 [37]	617	36/5.8	468/75.9 *	70 *	NA
Byun, 2020 [38]	18	18/100	10/55.6	64 (27–78)	22.6 (0.9–33.4)
Eshet, 2018 [39]	617	31/5	20/64.5	63 (35–80)	14.5 (6.5–20)
Eyada, 2021 [40]	171	85/49.7	62/72.9	61 (4) ^†^	2.5
Gleeson, 2023 [41]	1096	24/2.2	629/57.4 *	68 (59–75) *	NA
Grimmelmann, 2021 [42]	68	68/100	33/48.5	58 (20–90)	13 (8.1–22.8)
Hori, 2024 [43]	843	48/5.7	596/70.7 *	71 (26–91) *	NA
Jeun, 2023 [44]	76	76/100	43/56.6	60 (32–83)	45.9 (4.3–102.9)
Matsukane, 2023 [45]	1008	14/1.4	732/72.6 *	68 (14–89) *	NA
Nagao, 2024 [46]	1069	19/1.8	15/78.9	67 (57–78)	12.3 *
Nagpal, 2020 [47]	56	8/14.3	4/50	76 (61–88)	NA
Ngamphaiboon, 2021 [48]	414	11/2.7	287/69.3 *	63 (17–97) *	NA
Nizam, 2024 [49]	16	1/6.3	11/68.7 *	73 (47–79) *	NA
Owen, 2021 [50]	118	4/3.4	80/67.8 *	64 (30–89) *	21 *
Pollack, 2018 [51]	80	4/5	44/55 *	56 (25–89) *	14.3 *
Ruf, 2024 [52]	217	6/2.8	124/57.1 *	66 (23–89) *	NA
Satish, 2023 [53]	12,905	23/0.2	16/69.6	62 (8) ^†^	NA
Shah, 2019 [54]	1672	21/1.3	NA	NA	NA
Shi, 2022 [55]	1905	20/1	1442/75.7 *	63 (56–68) *	NA
Shirwaikar, 2021 [56]	11,165	248/2.2	NA	NA	NA
Townsend, 2023 [36]	6450	105/1.6	59/56.2	64 (24–88)	22.1 (1.2–100.6)
Tsang, 2019 [57]	538	10/1.9	9/90	62 (43–79)	NA
Wei, 2024 [10]	2829	10/0.4	8/80	63 (40–88)	NA
Zhang, 2023 [58]	2477	25/1	14/56	65	NA

^$^ All are retrospective studies. * Data represents all patients recruited in each study. ^†^ Mean (standard deviation). Abbreviations: ICI-PI = immune checkpoint inhibitor-induced pancreatic injury; IQR = interquartile range; NA = not available.

**Table 2 cancers-17-01080-t002:** Disease site and stage of patients included in the analysis.

Study ^$^	Tumor Sites (n/%)					Disease Stage (n/%)		Median Time of Onset, Days (Range)	Median Time to Resolution, Days (Range)	Patients with Elevated Lipase Levels (n/%)
	Skin	GI	Thoracic/ H&N	GU/Gynae	Others	Metastatic	Localized			
Abu-Sbeih, 2019 [11]	30/36.6	5/6.1	11/13.4	24/29.3	12/14.6	72/87.8	10/12.2	NA	Steroid-55(11) ^†^ IVF-55(51) ^†^	41/50
Akazawa, 2014 [37]	NA	NA	NA	NA	NA	36/100	0/0	105.5	NA	NA
Byun, 2020 [38]	5/27.8	2/11.1	1/5.6	8/44.4	2/11.1	16/88.9	2/11.1	109.5	NA	4/22
Eshet, 2018 [39]	21/67.7	0	10/32.3	0	0	NA	NA	270 ^#^	NA	NA
Eyada, 2021 [40]	10/12.8	0	10/11.8	40/47	25/29.4	63/74.1	22/26	NA	NA	NA
Gleeson, 2023 [41]	288/26.3 *	27/2.5 *	223/20.3 *	185/16.9 *	74/6.7 *	NA	NA	NA	NA	NA
Grimmelmann, 2021 [42]	68/100	0	0	0	0	68/100	0	135	NA	68/100
Hori, 2024 [43]	89/10.6 *	164/19.5 *	429/50.9 *	136/16 *	25/3 *	48/100	0	NA	NA	48/100
Jeun, 2023 [44]	23/30.2	0/0	12/15.8	11/14.5	30/39.5	NA	NA	86.1	NA	20/26
Matsukane, 2023 [45]	147/14.6 *	178/17.7 *	494/49 *	166/16.5 *	23/2 *	NA	NA	66	NA	NA
Nagao, 2024 [46]	4/21.1	1/5.3	4/21.1	10/52.5	0	335/31.4 *	734/68.6 *	92 (19–706)	NA	19/100
Nagpal, 2020 [47]	0	0	2/25	5/62.5	1/12.5	NA	NA	130.5 (1–434)	NA	8/100
Ngamphaiboon, 2021 [48]	33/8 *	66/15.9 *	235/56.8 *	60/14.5 *	20/5 *	408/98.6 *	6/1.4 *	30 (14–254)	84 (1–707)	NA
Nizam, 2024 [49]	0	0	0	1/100	0	1/100	0	NA	NA	NA
Owen, 2021 [50]	4/100	0	0	0	0	107/90.7 *	11/9.3 *	NA	NA	NA
Pollack, 2018 [51]	4/100	0	0	0	0	4/100	0	NA	NA	4/100
Ruf, 2024 [52]	206/94.9 *	3/1.4 *	2/0.9 *	6/2.8 *	0 *	NA	NA	63	NA	NA
Satish, 2023 [53]	8/34.8	2/8.7	4/17.4	6/26.1	3/13	NA	NA	390 (252–578) ^#^	NA	15/65
Shah, 2019 [54]	NA	NA	NA	NA	NA	NA	NA	NA	NA	17/81
Shi, 2022 [55]	0	0	20/100	0	0	1488/78.1 *	417/21.9 *	92.5	NA	NA
Shirwaikar, 2021 [56]	NA	NA	NA	NA	NA	NA	NA	NA	NA	248/100
Townsend, 2023 [36]	33/31.4	5/4.8	18/17.2	12/11.4	37/35.2	NA	NA	84	NA	105/100
Tsang, 2019 [57]	10/100	0	0	0	0	10/100	0	175	NA	NA
Wei, 2024 [10]	3/30	1/10	3/30	3/30	0	NA	NA	295.5	NA	1/10
Zhang, 2023 [58]	8/32	1/4	8/32	8/32	0	NA	NA	83.3	NA	NA

^$^ All are retrospective studies. * Data represent all patients recruited in each study. ^†^ Mean (standard deviation). ^#^ Time of onset of long-term complications. Abbreviations: H&N = head and neck; GI = gastrointestinal; GU = genitourinary; Gynae = gynecological; NA = not available; IVF = intravenous fluid. Incidence and grade of ICI-PI.

**Table 3 cancers-17-01080-t003:** Chronic complications of ICI-induced pancreatic injury.

	Number of Events (%)	95% CI
Chronic pancreatic complications	215 (63.45%)	29.03–91.56%
Diabetes mellitus	183 (89.45%)	61.88–100.0%
Insulin use	18 (80.07%)	24.49–100.0%
Oral hypoglycemic use	4 (19.93%)	0.0–75.51%
Exocrine pancreatic insufficiency	32 (10.55%)	0.0–38.12%
Pancrelipase use	25 (50.42%)	0.0–100.0%

## Data Availability

All data generated or analyzed during this study are included in this published article and its Appendix A.

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
