# Peer review of "Immune Checkpoint Inhibitor-Induced Pancreatic Injury (ICI-PI) in Adult Cancer Patients: A Systematic Review and Meta-Analysis"

_cancers, 2025, doi:10.3390/cancers17071080_

Round 1
Reviewer 1 Report
Comments and Suggestions for Authors
Immune checkpoint inhibitor-induced pancreatic injury (ICI-PI) represent the very interesting and not yet enough evaluated pathology.The incidence found in this analysis is surprisungly high.This is probably beacause the simple hiperlipasemia inclusion, which is controversial.We observe it a lot after ERCP procedure not considering it pathology, but a transient irritation with no longterm sequele.The pathogenesis is different, but the outcome may be similar.Is diabetes and exocrine pancreatic insufficiency directly due to Immune checkpoint inhibitor is not clear.No information was provided about the number of pancreatic cancer in the evaluated groups, which is connected to diabetes in up to 80%, again with no clear pathogenesis.In some cases the diabetes was probable to the prexisting , diagnosed or no type 2 diabetes.It would be also interesting to know how was the exocrine pancreatic insufficiency diagnosed?With elastase-1 test or breath tests?.Many other than reported drugs used in malignancies have diabetogenic effect and many other cancers are connected to paraneoplastc diabetes.But generally addressing his issue is of important clinical value.
Author Response
- Hyperlipasemia as an Inclusion Criterion: We acknowledge the reviewer’s concern regarding the high incidence of ICI-PI in this analysis and agree that it can be attributed to the inclusion of hyperlipasemia as one of the criteria (lines 285-290). While hyperlipasemia can occur transiently as a result of procedural irritation, especially following ERCP, we believe its inclusion is justified based on the broad definition of ICI-PI used in current oncology clinical practice. Our decision was supported by prior meta-analyses which showed that hyperlipasemia is commonly seen in ICI-related pancreatic events (71% of cases). Our analysis specifically includes cancer patients who have been exposed to ICI treatment, which further strengthens the argument that hyperlipasemia observed in this cohort is related to ICI drugs.
- ICI-induced Diabetes and Exocrine Pancreatic Insufficiency (EPI) direct association with ICI agents: We agreed with reviewer’s comment about the uncertainty regarding the direct association of diabetes, EPI and ICI therapy, and suggested that these could be related to other paraneoplastic or non-ICI drugs used in cancers patients. In this analysis using the patient selection criteria (detailed in Section 2.2, page 3), we hope to maintain the assumption that both diabetes and EPI are related to ICI therapy. However, the retrospective nature of the studies limits our ability to draw definitive conclusions about the sole causal link between ICI agents with these pancreatic conditions (lines 290-292). Furthermore, we have highlighted the potential diabetogenic effect of corticosteroids used in managing immune-related adverse events, as they can contribute to the observed ICI-induced diabetes cases, and remain as potential area for future research (lines 359-361).
- Pancreatic Cancers and Diabetes: Upon revisiting the data, we found that only four cases of pancreatic cancers were included in our analysis (Ngamphaiboon et al., 2021). Due to the small number of cases, it is not possible to establish a clear causal link between ICI-PI and pancreatic cancer in our analysis. We have added this information in the revised manuscript (lines 295-298).
- Pre-existing Diabetes: Regarding the relationship between ICI and ICI-induced diabetes, we recognize that a proportion of diabetes cases in our cohort may be pre-existing or undiagnosed Type 2 diabetes. As mentioned in line 358, almost 15% of our study population had a history of diabetes prior to ICI therapy. The retrospective nature of this study limits our ability to definitively separate pre-existing diabetes from newly onset ICI-DM.
- EPI Diagnosis: We agree with the reviewer’s suggestion and have revised the manuscript to include this information (lines 222-224). Our analysis acknowledges variability in the diagnostic methods used for EPI across the studies. We have also included this as a study limitation (lines 352-353).
Reviewer 2 Report
Comments and Suggestions for Authors
The manuscript by Cha Len Lee and co-authors describes one of the side effects of current immunotherapy, pancreatic injury induced by immune checkpoint inhibitors. This condition is important because it could provoke functional pancreatic deficiency and diabetes. This paper could be very interesting for the readers of Cancer journal, but the data should be presented in a more readable form.
For example, figures 2 - 5 are hard to read. The summary table should be introduced to allow conclusions to be drawn about the main consequences of immunotherapy related pancreatic injury.
The typographical errors and misprints throughout the text should be corrected.
Author Response
- Figure Quality: We apologize for the suboptimal quality of figures in the initial submission. We believe that the image quality was compromised during the file compression. To address this, we have prepared high-resolution PDF versions of the figures and will provide them directly to the editorial office.
- Summary Table: As suggested, we have added a new Table 3 in Section 3.3 (page 9). This table summarizes the number and proportion of chronic complications associated with ICI-PI, as well as their management strategies. We believe this addition will enhance the clarity of the manuscript and provide an overview of the adverse outcomes related to ICI-PI.
- Typographical and Syntax Errors: We sincerely apologized for this. We have thoroughly reviewed the manuscript and corrected the identified typographical errors and syntax issues.
Round 2
Reviewer 2 Report
Comments and Suggestions for Authors
The necessary corrections have been made. The manuscript has been improved.